# Metabolic and Inflammatory Adipokine Profiles in PCOS: A Focus on Adiposity, Insulin Resistance, and Atherogenic Risk

**DOI:** 10.3390/ijms26199702

**Published:** 2025-10-05

**Authors:** Daniela Koleva-Tyutyundzhieva, Maria Ilieva-Gerova, Tanya Deneva, Maria Orbetzova

**Affiliations:** 1Department of Endocrinology and Metabolic Diseases, Faculty of Medicine, Medical University of Plovdiv, 4002 Plovdiv, Bulgaria; maria.ilieva@mu-plovdiv.bg (M.I.-G.); maria.orbetzova@mu-plovdiv.bg (M.O.); 2Department of Clinical Laboratory, Faculty of Medicine, Medical University of Plovdiv, 4002 Plovdiv, Bulgaria; tanya.deneva@mu-plovdiv.bg

**Keywords:** polycystic ovary syndrome, adiposity, insulin resistance, adipokines, atherogenic indices, metabolic dysfunction, inflammation

## Abstract

Polycystic ovary syndrome (PCOS) is a prevalent endocrine disorder connected with insulin resistance (IR), low-grade inflammation, dyslipidemia, and altered adipokine secretion. We evaluated serum levels of leptin, adiponectin, visfatin, resistin, IL-6, and TNF-α in 150 women with PCOS, stratified by IR status (IR, *n* = 76; non-IR, *n* = 74), and examined their associations with anthropometric, metabolic, hormonal, inflammatory, and atherogenic parameters. Anthropometric data included body weight, height, BMI, waist circumference, and waist-to-height ratio (WHtR), while IR was assessed using HOMA-IR and the Matsuda index. Serum adipokines were measured using ELISA, and lipid parameters and atherogenic indices—including non-HDL cholesterol, AIP, leptin/adiponectin, and adiponectin/resistin ratios—were calculated. Women with IR had higher levels of leptin, visfatin, resistin, and TNF-α, and lower levels of adiponectin. Leptin correlated positively with weight, WHtR, HOMA-IR, and atherogenic indices. Adiponectin showed the strongest and most consistent associations with anthropometric indices, HOMA-IR, and the Matsuda index. Resistin was linked to IR indices and IL-6, and visfatin correlated negatively with HDL-C and insulin sensitivity. In a multivariate general linear model, WHtR, but not HOMA-IR, remained independently associated with higher leptin levels and with atherogenic indices. These findings suggest that in PCOS, central adiposity rather than IR explains a substantial part of the adverse adipokine and inflammatory profile, thereby contributing to elevated cardiometabolic risk and highlighting the need for targeted treatment strategies.

## 1. Introduction

Polycystic ovary syndrome (PCOS) is a multifactorial endocrine disorder affecting approximately 10–15% of women at reproductive age, depending on the diagnostic criteria, and is one of the most common ovarian pathologies worldwide [1]. It is classically defined by the presence of oligo- or anovulation, clinical and/or biochemical hyperandrogenism, and polycystic ovarian morphology, with the Rotterdam criteria further classifying PCOS into four phenotypes (A–D) based on combinations of these features, which differ in reproductive, metabolic, and cardiovascular risk profiles [2].

While historically regarded as a reproductive disorder, PCOS is now recognized as a complex metabolic condition, characterized by insulin resistance (IR), dyslipidemia, central obesity, and chronic low-grade inflammation [3,4,5].

Insulin resistance occurs in up to 70% of women with PCOS and is more prevalent in those with higher adiposity, although it can also be observed in lean phenotypes. It is known to play a pivotal role in the development of both reproductive and metabolic abnormalities [6]. IR contributes to compensatory hyperinsulinemia, which stimulates ovarian androgen production, disrupts folliculogenesis, and alters adipose tissue signaling [7]. Dysfunctional adipose tissue, characterized by the altered secretion of adipokines—bioactive cytokines produced by adipocytes—has emerged as a key component of this metabolic–inflammatory axis, influencing insulin sensitivity, lipid metabolism, and immune responses [8,9].

Among adipokines, leptin, adiponectin, visfatin, and resistin have been most extensively studied due to their distinct insulin-sensitizing or proinflammatory actions. Leptin and resistin levels are generally elevated in insulin-resistant states and show positive correlations with adiposity, proinflammatory cytokines such as TNF-α and IL-6, and cardiometabolic risk markers [10,11,12]. Adiponectin, an insulin-sensitizing and anti-inflammatory adipokine, is typically reduced in PCOS and inversely related to IR and dyslipidemia [13,14]. Visfatin has been reported to exert both insulin-mimetic and proinflammatory effects, although findings remain inconsistent [15]. Alterations in adipokine and cytokine profiles, including TNF-α and IL-6, may not only reflect underlying metabolic dysfunction but also contribute to systemic inflammation and reproductive impairment in PCOS [16].

Beyond IR, PCOS is frequently associated with atherogenic dyslipidemia, characterized by elevated triglycerides (TGs), reduced high-density lipoprotein cholesterol (HDL-C), increased small dense low-density lipoprotein (sdLDL) particles, and unfavorable lipid ratios such as TG/HDL-C and LDL-C/HDL-C [17,18]. These alterations can be summarized by composite indices such as the Atherogenic Index of Plasma (AIP), a sensitive marker of cardiovascular risk [19]. Evidence suggests that certain adipokines may directly influence lipid metabolism, thereby linking adipose tissue dysfunction to atherogenesis [20]. In this context, assessing adipokine ratios, such as the leptin-to-adiponectin (L/A) and adiponectin-to-resistin (A/R) ratios, may provide more robust markers of metabolic dysfunction than single adipokine measurements, as they reflect the net balance between proinflammatory and insulin-sensitizing signals [21,22,23]. In parallel, proinflammatory cytokines, particularly TNF-α and IL-6, play a central role in amplifying insulin resistance by disrupting insulin receptor signaling pathways and promoting dyslipidemia [24]. Their interplay with adipokines may create a feed-forward loop of inflammation and metabolic impairment in PCOS.

However, few studies have simultaneously examined adipokines, IR, and lipid-related atherogenic markers in PCOS populations. Moreover, the extent to which insulin resistance drives adipokine–lipid associations remains unclear.

Therefore, this study aimed to compare serum concentrations of leptin, adiponectin, visfatin, resistin, TNF-α, and IL-6 in women with PCOS stratified by insulin resistance status, and to explore their relationships with anthropometric, metabolic, hormonal, inflammatory, and atherogenic parameters. Elucidating these interactions may help clarify mechanisms underlying cardiometabolic risk in PCOS and identify potential targets for individualized metabolic intervention.

## 2. Results

### 2.1. Baseline Characteristics of the PCOS Cohort

Analysis of clinical characteristics revealed notable differences in BMI distribution between women with PCOS who have and do not have insulin resistance (IR). Among non-IR women, 67.4% had a BMI within the normal range (18.5–24.9 kg/m^2^), 14.0% were overweight (25.0–29.9 kg/m^2^), 14.0% were classified as class I obesity (30.0–34.9 kg/m^2^), and 4.7% were classified as class II obesity (35.0–39.9 kg/m^2^). In contrast, women with IR showed a shift toward higher BMI categories: 33.3% had a BMI of 18.5–24.9 kg/m^2^, 18.2% were overweight, 39.4% were classified as class I obesity, and 9.1% were classified as class II obesity. This difference in BMI distribution between the two groups was statistically significant (Chi-square = 9.73, *p* = 0.021).

Regarding menstrual cycle disturbances, oligo-amenorrhea was the predominant abnormality in both groups, affecting 79.1% of non-IR and 75.0% of IR women. However, the prevalence of menstrual irregularities did not differ significantly between groups (Chi-square = 0.173, *p* > 0.05). Similarly, the prevalence of hirsutism was 62.8% in non-IR women and 51.5% in IR women, with no significant difference observed (Chi-square = 0.97, *p* > 0.05). Acne was reported in 27.8% of non-IR women and 27.3% of IR women, again without significant group differences (Chi-square = 0.004, *p* > 0.05). Polycystic ovarian morphology (PCO) on ultrasound was present in 57.5% of non-IR and 65.5% of IR women, with no statistically significant difference between groups (Chi-square = 0.45, *p* > 0.05).

Phenotype distribution was as follows: phenotype A—40.8% (*n* = 61), phenotype B—35.5% (*n* = 53), phenotype C—17.1% (*n* = 26), and phenotype D—6.6% (*n* = 10).

### 2.2. Age and Anthropometric Parameters

Table 1 summarizes the age and anthropometric parameters of women with and without IR and PCOS. Both groups were similar in age and height. Compared to non-IR women, those with IR had significantly higher body weight, BMI, waist circumference, and waist-to-height ratio (WHtR).

### 2.3. Adipokine Profiles

Adipokine profiles differed significantly between groups (Table 2). Compared with non-IR women, those with IR exhibited higher visfatin (*p* < 0.05), leptin (*p* < 0.01), log_10_-transformed resistin (*p* < 0.05) values, and lower adiponectin levels (*p* < 0.01). Serum TNF-α levels were significantly higher in the IR group compared with the non-IR group (*p* < 0.05, Mann–Whitney test). IL-6 levels were similar between groups (*p* > 0.05).

### 2.4. Glucose and Insulin Dynamics

During OGTT, IR women exhibited higher GLU and IRI concentrations compared to non-IR participants: GLU 0′-5.21 ± 0.42 vs. 4.62 ± 0.36 mmol/L, GLU 60′-8.42 ± 1.21 vs. 5.84 ± 0.98 mmol/L, GLU 120′-6.94 ± 1.15 vs. 5.14 ± 0.82 mmol/L(*p* < 0.001); IRI 0′- 12.80 ± 8.7 vs. 6.21 ± 3.2 μIU/mL, IRI 60′-114.02 ± 45.6 vs. 47.41 ± 21.3 μIU/mL, and IRI 120′-88.50 ± 38.1 vs. 27.54 ± 18.7 μIU/mL (*p* < 0.001) (Figure 1 and Figure 2). Despite these differences, mean glucose values in both groups remained below diabetic thresholds.

In the IR cohort, HOMA-IR was significantly elevated (3.11 ± 1.77 vs. 1.28 ± 0.48; *p* < 0.001), accompanied by a markedly lower Matsuda index (3.97 ± 1.73 vs. 11.77 ± 5.55; *p* < 0.001) (Figure 3 and Figure 4). These findings confirm the validity of the insulin resistance classification applied in this study.

### 2.5. Lipid Profile and Atherogenic Indices

Table 3 presents lipid parameters and atherogenic indices in both groups. Compared to non-IR women, the IR group demonstrated significantly higher values of triglycerides (TG) (*p* < 0.001), Atherogenic Index of Plasma (AIP) (*p* < 0.001), leptin-to-adiponectin ratio (L/A) (*p* < 0.001), and adiponectin-to-log resistin ratio (A/R) (*p* = 0.001). No significant differences concerning total cholesterol (TC), HDL-C, LDL-C, or non-HDL-C were observed (Table 3).

### 2.6. Hormonal Parameters

Table 4 presents hormonal parameters in PCOS women according to the insulin resistance status. Total testosterone was significantly higher in the IR group compared to non-IR women (*p* < 0.05). No significant differences were observed between the groups for SHBG, androstenedione, DHEA-S, or FAI (Table 4).

### 2.7. Correlation Analyses

#### 2.7.1. Leptin

Serum leptin levels showed significant positive correlations with several anthropometric and metabolic parameters. Strong associations were observed with body weight (*r* = 0.732, *p* < 0.001; Figure 5), BMI (*r* = 0.694, *p* < 0.001), waist circumference (*r* = 0.679, *p* < 0.001), and waist-to-height ratio (*r* = 0.622, *p* < 0.001). Leptin also correlated moderately with HOMA-IR (*r* = 0.409, *p* < 0.001; Figure 6) and showed a negative correlation with the Matsuda insulin sensitivity index (*r* = −0.423, *p* < 0.001; Figure 7). All these associations remained significant after correction for multiple testing (FDR method).

In addition, leptin showed weaker correlations with lipid-related markers, including a positive association with the AIP (*r* = 0.246, unadjusted *p* < 0.05), non-HDL-C (*r* = 0.288, unadjusted *p* < 0.05), and a negative association with the A/R (*r* = −0.276, unadjusted *p* < 0.05). However, these findings did not remain statistically significant after multiple-testing adjustment and should be considered exploratory.

Linear regression analysis confirmed body weight as the strongest determinant of serum leptin, accounting for 54% of its variance (R = 0.732, R^2^ = 0.536, *p* < 0.001, F = 73.96). HOMA-IR explained 17% of leptin variability (R = 0.409, R^2^ = 0.167, *p* < 0.001, F = 12.85). Together, these results indicate that leptin levels primarily reflect adiposity but also capture insulin resistance to a lesser extent.

Serum leptin concentrations demonstrated an inverse relationship with SHBG levels (*r* = −0.403, *p* = 0.037). However, following correction for multiple comparisons using the FDR method, this association did not reach statistical significance, indicating that the observed correlation should be interpreted cautiously.

#### 2.7.2. Adiponectin

Adiponectin demonstrated significant inverse correlations with anthropometric and metabolic parameters. Lower adiponectin concentrations were associated with higher body weight (*r* = −0.385, *p* = 0.001), BMI (*r* = −0.361, *p* = 0.003), waist circumference (*r* = −0.411, *p* = 0.001), and WHtR (*r* = −0.378, *p* = 0.002). Negative associations were also observed with fasting insulin (*r* = −0.332, *p* = 0.006), IRI at 60 min (*r* = −0.288, *p* = 0.027), IRI at 120 min (*r* = −0.259, *p* = 0.047), HOMA-IR (*r* = −0.308, *p* = 0.012), triglycerides (*r* = −0.409, *p* = 0.010), and AIP (*r* = −0.422, *p* < 0.001). Conversely, adiponectin correlated positively with the calculated insulin sensitivity Matsuda index (*r* = 0.423, *p* < 0.001). After adjusting for multiple testing (FDR method), the strongest and most consistent associations were found with anthropometric indices (BMI, waist circumference, WHtR), HOMA-IR, and the Matsuda index, while correlations with post-load insulin values and triglycerides did not remain statistically significant and should be considered exploratory.

#### 2.7.3. Visfatin

Visfatin showed a significant inverse correlation with HDL-C (*r* = −0.376, *p* = 0.024). In addition, a borderline negative association with the Matsuda insulin sensitivity index was observed (Kendall’s *τ* = −0.226, *p* = 0.051). However, this latter finding did not reach statistical significance after adjustment for multiple testing and should be interpreted as exploratory.

#### 2.7.4. Resistin

Study results showed positive relationships between resistin and the following glucose and insulin metabolism parameters: GLU 0′ (r = 0.278, *p* = 0.024), IRI 120′ (r = 0.315, *p* = 0.015), and HOMA-IR (*r* = 0.272, *p* = 0.027). Additionally, resistin was found to be inversely correlated with the Matsuda index (*r* = −0.243, *p* = 0.050). Furthermore, it showed a positive correlation with IL-6 (*r =* 0.270, *p* = 0.028). After adjustment for multiple testing (FDR method), only the associations with post-load insulin and IL-6 remained significant, while correlations with fasting glucose, HOMA-IR, and the Matsuda index were no longer statistically significant and warrant cautious interpretation.

### 2.8. Multivariable Analysis

In the multivariate general linear model (Table 5), WHtR—but not HOMA-IR—was independently associated with serum leptin (WHtR: F = 6.98, *p* = 0.013; HOMA-IR: F = 1.50, *p* = 0.230).

Similarly, WHtR predicted the leptin/adiponectin ratio (L/A: F = 10.74, *p* = 0.003), AIP (AIP: F = 10.97, *p* = 0.002), and non-HDL-C (F = 6.71, *p* = 0.014) (Table 6).

HOMA-IR did not show significant independent associations with the tested adipokines or most atherogenic indices in these age-adjusted models (all *p* > 0.05).

## 3. Discussion

Our study encompassed women representing all four Rotterdam PCOS phenotypes, with the majority classified as phenotypes A and B, enhancing the generalizability of our findings across the PCOS spectrum. Importantly, exclusions for smoking and the use of lipid- or insulin-modifying agents minimized potential confounding factors.

This study provides evidence that IR is associated with alterations in the adipokine profile in women with PCOS, which may contribute to adverse metabolic and cardiovascular risk. However, given the higher BMI and central fat in the IR group, the observed differences cannot be disentangled from the effects of adipose tissue. Our multivariable analyses revealed that WHtR, but not HOMA-IR, was independently associated with serum leptin, the L/A, AIP, and non-HDL-C, highlighting central adiposity as a key driver of both adipokine alterations and cardiometabolic risk. HOMA-IR did not show significant independent associations with adipokines or most atherogenic indices, suggesting that insulin resistance may not fully explain the observed metabolic dysregulation.

Elevated leptin levels—particularly in women with higher WHtR—likely reflect leptin resistance and correlated strongly with anthropometric indices, non-HDL-C, and AIP, supporting its potential role as an independent predictor of cardiometabolic risk. Adiponectin concentrations were lower in women with increased central adiposity and inversely associated with L/A, TG, and AIP, reinforcing its anti-atherogenic properties and sensitivity to fat distribution.

Visfatin and resistin levels were elevated in women with increased central adiposity, suggesting contributions to impaired insulin sensitivity, low-grade inflammation, and dyslipidemia. The positive correlation between resistin and IL-6 further supports a link between adipokine dysregulation and proinflammatory signaling. TNF-α levels were higher in insulin-resistant women, reflecting the chronic inflammatory milieu characteristic of PCOS. In contrast, IL-6 was not independently associated with anthropometric or metabolic parameters, suggesting a more complex regulation of this cytokine.

Taken together, these findings suggest that central adiposity, rather than insulin resistance alone, is a major driver of the metabolically and inflammatory dysregulated phenotype in PCOS, potentially amplifying reproductive and cardiometabolic abnormalities.

Our results of elevated leptin levels in IR women with PCOS are consistent with numerous previous reports [25,26]. Hyperleptinemia in this context likely reflects a state of leptin resistance, characterized by impaired hypothalamic feedback despite high circulating levels [27]. The strong correlations between leptin and anthropometric indices, HOMA-IR, and the Matsuda index in our cohort mirror results from Panidis et al. [28] and Nyangasa et al. [29], who also noted leptin’s positive associations with markers of central obesity. Additionally, Chakrabarti et al. observed that leptin concentrations were more than two-fold higher in IR PCOS patients compared with controls and showed strong positive correlations with circulating insulin [30]. Similarly, Jahromi et al. demonstrated robust associations between leptin and HOMA-IR, QUICKI, body weight, and BMI in infertile women with PCOS, identifying HOMA-IR as the most sensitive marker of insulin resistance [31].

In agreement with our data, Daghestani et al. reported positive correlations between leptin and BMI, WHR, TC, LDL-C, and TG, and an inverse correlation with HDL-C [32]. In our cohort, BMI was the strongest determinant of leptin levels, although circulating insulin also contributed, suggesting that both adiposity and insulin dynamics play important roles in modulating adipokine concentrations. Moreover, the observed positive association between leptin and both non-HDL-C and the AIP is consistent with accumulating evidence suggesting that leptin may serve as an independent predictor of cardiometabolic risk in women with PCOS [12,33]. Lee et al. reported that TG/HDL-C was significantly higher in PCOS and served as a useful surrogate marker of insulin IR and cardiometabolic risk [34], while Demirci et al. showed that AIP was elevated in PCOS and independently predicted by both PCOS status and HOMA-IR [35].

Experimental data provide further insight into the reproductive interface. In vitro studies on human theca and granulosa cells show that leptin can inhibit IGF-I-mediated augmentation of LH-induced androgen and progesterone synthesis at concentrations typical of obesity, suggesting dysregulated ovarian leptin signaling in PCOS [36,37].

Adiponectin, in contrast, exhibited significantly lower concentrations in individuals with IR and was positively associated with the Matsuda index, in agreement with several meta-analyses [38,39]. Its inverse correlations with TG and AIP reinforce its recognized anti-atherogenic properties, possibly mediated via AMPK activation, enhanced fatty acid oxidation, and suppression of foam cell formation [20,40]. While low adiponectin is a consistent hallmark of PCOS across most populations [41], some studies have noted that its predictive utility varies by phenotype, obesity status, and ethnicity [42,43].

A systematic review and meta-analysis by Toulis et al. [38] reported that, after controlling for BMI-related effects, adiponectin levels were significantly lower in women with PCOS compared to non-PCOS controls—a finding present in both lean and obese phenotypes. Similarly, Patil et al. [44] observed that serum adiponectin levels were decreased in women with PCOS and inversely associated with BMI, TC, and TG, suggesting potential diagnostic and prognostic value.

The association between adiponectin and IR in PCOS has also been highlighted by Shirazi et al. [45], who demonstrated a significant correlation between insulin levels and the free androgen index independent of obesity. This underscores the central role of IR in PCOS pathophysiology. Phenotypic variations further modulate adiponectin levels, as shown by Barrea et al. [46], who reported that obese PCOS patients exhibited a greater prevalence of metabolic and reproductive disturbances compared to lean counterparts, suggesting that obesity and PCOS phenotype influence circulating adiponectin concentrations. In contrast, genetic studies, such as those by Nowak et al. [47], found that adiponectin gene polymorphisms (e.g., rs17300539) were not significantly associated with metabolic syndrome in PCOS, indicating that environmental and metabolic factors may play a more prominent role than genetics in determining adiponectin levels in this population.

Visfatin’s role in PCOS remains controversial. Initially described as an insulin-mimetic adipokine (nicotinamide phosphoribosyltransferase, NAMPT) [48], subsequent research has highlighted its proinflammatory actions through NF-κB activation and upregulation of IL-6 [49]. In our study, increased visfatin levels in IR women, along with negative associations with HDL-C and the Matsuda index, suggest that visfatin may contribute to both impaired insulin sensitivity and atherogenic dyslipidemia. Similar associations were reported by Kowalska et al. [50]. In contrast, other studies have found no significant differences in visfatin levels between PCOS phenotypes [51], underscoring the need for stratified analyses based on metabolic status.

Building upon these observations, evidence from a comprehensive meta-analysis encompassing 1,341 women (695 with PCOS and 646 clinically healthy controls) provides further insight into visfatin dynamics in PCOS [52]. The primary objective of this study was to evaluate serum visfatin levels across the two cohorts and to perform a comparative analysis. The results unequivocally demonstrated significantly elevated serum visfatin concentrations in women with PCOS relative to controls. Notably, stratified and univariate analyses revealed no significant associations between heightened visfatin levels and BMI, HOMA-IR, or testosterone concentrations [52]. These findings suggest that elevated circulating visfatin may constitute a distinct intragroup characteristic of PCOS, highlighting the potential utility of this adipokine as a diagnostic biomarker for the syndrome.

El-Said et al. reported significantly elevated plasma visfatin concentrations in a cohort of IR women with PCOS (72.94 ± 33.3 ng/mL) compared with clinically healthy controls (54.69 ± 31.5 ng/mL, P = 0.039). Within the PCOS group, visfatin demonstrated positive correlations with BMI, waist circumference, HOMA-IR, and free androgen index (FAI), and inverse correlations with luteinizing hormone (LH), total testosterone, and sex hormone-binding globulin (SHBG). Across the overall study population, plasma visfatin concentrations were inversely associated with HDL-C (r = −0.349, P = 0.013), highlighting a potential link with atherogenic risk [53]. Interestingly, in contrast to these observations, Gen et al. reported a positive correlation between plasma visfatin and HDL-C in a cohort of women with PCOS who exhibited normal body weight, suggesting that the relationship between visfatin and lipid metabolism may be modulated by metabolic status [54]. Taken together, these findings underscore the complexity of visfatin’s role in PCOS and emphasize the importance of stratified analyses according to insulin resistance and body composition.

Resistin, although less studied in PCOS, has shown positive correlations with fasting/postprandial insulin levels and an inverse relationship with the Matsuda index. These results are consistent with findings from Estienne et al. [55] and Bril et al. [56], who linked elevated resistin to systemic inflammation, IR, and endothelial dysfunction in PCOS. Similarly, Lewandowski et al. [57] reported that serum resistin levels were significantly higher in PCOS patients compared to BMI-matched controls and correlated with HOMA-IR and markers of subclinical inflammation, supporting a role in metabolic dysregulation. In a study by Yildiz et al. [58], resistin was found to be positively associated with proinflammatory cytokines such as TNF-α and CRP, suggesting its involvement in the low-grade chronic inflammation characteristic of PCOS, which is in line with our finding of a positive correlation between resistin and IL-6. Mechanistically, resistin may impair insulin signaling by upregulating suppressor of cytokine signaling-3 (SOCS-3) and enhancing vascular inflammation [59]. Experimental data, however, indicate that resistin can also promote hepatic gluconeogenesis and reduce glucose uptake in adipocytes, thereby further exacerbating insulin resistance [57]. Collectively, these findings suggest that resistin not only reflects metabolic and inflammatory disturbances in PCOS but may also actively contribute to the pathophysiology of insulin resistance and cardiometabolic risk in this population.

TNF-α is considered to be a key proinflammatory cytokine implicated in the pathogenesis of insulin resistance and associated metabolic disturbances in PCOS. In the present study, serum TNF-α concentrations were significantly elevated in IR women with PCOS compared to their non-IR counterparts, reinforcing the involvement of TNF-α in the chronic, low-grade inflammatory milieu characteristic of the syndrome. Increased TNF-α may exacerbate insulin signaling defects, contribute to dyslipidemia, and disrupt ovarian function, underscoring its potential utility as both a biomarker and a therapeutic target in addressing the metabolic sequelae of PCOS [60].

Our data support the hypothesis that adipokine dysregulation in PCOS is closely linked to IR, adiposity, systemic inflammation, and lipid abnormalities, all of which synergistically contribute to an elevated cardiometabolic risk profile [61]. Notably, we extend prior knowledge by demonstrating that specific adipokines—particularly leptin and adiponectin—correlate with composite atherogenic indices such as AIP, which are seldom evaluated in PCOS research.

In recent years, interest has expanded toward adipokine-based indices such as the leptin-to-adiponectin (L/A) and adiponectin-to-resistin (A/R) ratios. Several studies have demonstrated that the L/A is a sensitive predictor of insulin resistance and cardiometabolic risk, exceeding the diagnostic utility of individual adipokines [21,22,23]. The combination of low adiponectin levels with elevated leptin levels reflects impaired adipokine signaling, which is associated with endothelial dysfunction and atherogenesis. In addition, the A/R has been increasingly recognized as an indicator of metabolic disturbances. Maitra et al. [23] reported that a reduction in the A/R ratio in women with PCOS was significantly correlated with higher BMI, dyslipidemia, and HOMA-IR, highlighting its prognostic value for cardiometabolic risk.

While the association between insulin resistance and altered adipokine levels in PCOS has been described previously, our multivariable analyses add novelty by integrating adipokine and inflammatory markers with anthropometric indices and lipid surrogates. Importantly, WHtR, rather than HOMA-IR, emerged as the independent predictor of L/A, AIP, non-HDL-C, and leptin levels. The marked elevation of the L/A and TNF-α concentrations, along with the decrease in A/R, despite the absence of differences in non-HDL-Ц, underscores the potential of adipokine–inflammatory profiling combined with central adiposity measures for early cardiometabolic risk assessment in young women with PCOS. Future studies should investigate how adipokine dysregulation varies across PCOS phenotypes and how it relates to reproductive outcomes, thereby further linking the metabolic and endocrine features of the syndrome.

From a clinical perspective, these results highlight the potential value of adipokine profiling in early risk stratification and tailored management of PCOS. Interventions targeting IR—such as metformin, inositol isomers, and GLP-1 receptor agonists—have been shown to modulate adipokine levels and improve metabolic and reproductive outcomes [62,63,64]. Likewise, lifestyle interventions focusing on weight reduction, physical activity, and dietary patterns with low glycemic load or anti-inflammatory potential have demonstrated beneficial effects on adipokine balance and cardiometabolic markers [65,66].

Nevertheless, certain limitations should be acknowledged. We performed multivariable analyses adjusted for age; however, residual confounding by adiposity cannot be completely ruled out. Although WHtR was included and remained an independent predictor, direct imaging of abdominal fat (DXA/CT/MRI) was not included in all models. Future studies should include direct measures of visceral fat and formal mediation analyses to disentangle the relative contributions of adiposity vs. insulin resistance. Although TNF-α was elevated in the IR group, only two inflammatory markers (IL-6 and TNF-α) were assessed, and hs-CRP was not available; thus, broader inflammatory profiling would be needed to substantiate these findings. Finally, while the inclusion of both leptin-to-adiponectin and adiponectin-to-resistin ratios strengthens the evaluation of cardiometabolic risk, the absence of ApoB measurements and the lack of difference in non-HDL cholesterol between groups restricts conclusions regarding lipid-related risk. We did not stratify metabolic and adipokine results by PCOS phenotype, which may limit the interpretation of phenotype-specific results. Future studies should address potential differences in adipokine and inflammatory profiles across PCOS phenotypes. The cross-sectional nature of the study precludes causal inference, and circulating adipokine concentrations may not fully reflect tissue-specific activity or receptor sensitivity. Moreover, we did not assess genetic variants in adipokine-related genes, which may influence individual responses to treatment. Future longitudinal and interventional studies are warranted to clarify the causal pathways linking adipokine changes with long-term cardiovascular and reproductive outcomes in PCOS.

## 4. Materials and Methods

### 4.1. Study Design and Population

This cross-sectional, observational study included 150 women aged 18–35 years, diagnosed with polycystic ovary syndrome (PCOS) according to the Rotterdam criteria (2003) [2]. Diagnosis required at least two of the following: oligo/anovulation, clinical and/or biochemical hyperandrogenism, and polycystic ovarian morphology on ultrasound, with exclusion of other pathologies (e.g., congenital adrenal hyperplasia, Cushing’s syndrome, androgen-secreting tumors, thyroid dysfunction, and hyperprolactinemia). Based on these criteria, participants were classified into four PCOS phenotypes (A–D).

Participants were recruited from the Clinic of Endocrinology and Metabolic Diseases at “Sv. Georgy” University Hospital of Plovdiv between June 2020 and June 2023. Exclusion criteria included type 2 diabetes (ruled out via 75 g OGTT), chronic inflammatory or autoimmune disorders, thyroid or adrenal disease, use of insulin-sensitizing or lipid-lowering agents (e.g., metformin and statins), hormonal contraceptives, pregnancy, and current smoking. None of the women included in the final cohort were smokers.

For analysis, participants were stratified into two groups based on insulin resistance (IR) status: 1. PCOS group without IR (*n* = 74), and 2. PCOS group with IR (*n* = 76), defined by HOMA-IR ≥ 2.5 [67].

### 4.2. Anthropometric and Clinical Measurements

Anthropometric evaluation included body weight, height, and waist circumference (W), measured using standard techniques. Waist circumference was determined after the act of expiration, measuring the area between the bottom edges of the ribs and the iliac crests. Body mass index (BMI) {weight (kg)/height^2^ (m^2^)} and waist-to-height ratio (WHtR) were calculated.

### 4.3. Biochemical, Hormonal, and Adipokine Assessment

Venous blood for laboratory tests was collected under standard conditions: early in the morning, after an overnight 12 h fast period, during the follicular phase of the menstrual cycle (2nd to 5th day after a spontaneously occurring menstrual cycle) or 7 days after gestagen-induced bleeding. A 75 g oral glucose tolerance test (OGTT) was performed with blood sampling at 0, 60, and 120 min for plasma glucose (GLU) and insulin (IRI) levels. Samples for the determination of GLU and IRI, lipid parameters, standard hormonal parameters, and adipokines were taken to the Central Clinic Laboratory, “Sv. Georgy” University Hospital of Plovdiv, Bulgaria.

Serum insulin levels were determined using a chemiluminescent immunoassay (CLIA) kit from Beckman Coulter, Inc. (Chaska, MN, USA; manufactured in Ireland). This sandwich immunoassay method showed the following characteristics: dilution recovery: 96–104%; sensitivity: 0.03 μIU/mL; intra-assay variation (CV): 2.0–4.2%; inter-assay variation (CV): 3.1–5.6%; specificity: no cross-reactivity with bilirubin (10 mg/dL), triglycerides (20.32 mmol/L), or C-peptide (20,000 pmol/L) was observed; reference range: 1.9–23.0 μIU/mL. Serum glucose levels were tested by a standard GOD-POD method.

Insulin resistance and sensitivity were evaluated using the following indices: HOMA-IR = (Fasting insulin [µU/mL] × Fasting glucose [mmol/L])/22.5; and the Matsuda index = 10,000/√[(Fasting glucose [mg/dL] × Fasting insulin [µU/mL]) × (Mean OGTT glucose × Mean OGTT insulin)].

Serum lipids were measured enzymatically. Concentrations of total cholesterol (TC) were determined by ChOD, PAP; those of TG—using GPO, PAP, and HDL-C—through MgSO4-dextran SO4 precipitation. Analyses were performed using reagents from Schneiders Analysers (Schneiders Medizintechnik, Zwolle, Netherlands) and a Delta Kone autoanalyser (Kone Instruments, Espoo, Finland). LDL-C was calculated using the Friedewald formula, non-HDL-C was defined as TC–HDL-C, and the Atherogenic Index of Plasma (AIP) was calculated as log(TG/HDL-C).

Serum total testosterone (TT) levels were measured using chemiluminescent immunoassay (CLIA) kit (Beckman Coulter, Inc., Brea, CA, USA), with assay characteristics: dilution recovery: 96–115%; sensitivity: 0.35 ng/mL; intra-assay variation: CV 1.67–3.93%; inter-assay variation: CV 4.22–7.08%; specificity: no cross-reactivity was observed with bilirubin (10 ng/dL), triglycerides (20.32 mmol/L), hemoglobin (10 g/L), or albumin (55–85 g/L); reference ranges for women (21–73 years): 0.1–0.75 ng/mL. Serum androstenedione (A4) levels were measured using chemiluminescent immunoassay (CLIA) kit (catalog no. L2KAO2; Siemens Healthcare Diagnostics, Inc., Tarrytown, NY, USA) with assay characteristics: dilution recovery: 94–108%; sensitivity: 1.0 nmol/L; intra-assay variation: CV 6.2–15.1%; total imprecision: CV 8.5–17.8%; specificity: no cross-reactivity was observed with cholesterol (1000 ng/mL) or corticosterone (1000 ng/mL); reference ranges for women (follicular phase): 0.75–3.1 ng/mL. Serum dehydroepiandrosterone sulfate (DHEA-S) levels were measured using chemiluminescent immunoassay (CLIA) kit (Beckman Coulter, Inc., Brea, CA, USA) with the following assay characteristics: dilution recovery: 94.8–116.2%; sensitivity: < 2 μg/dL; intra-assay variation: CV 1.6–8.3%; inter-assay variation: CV 4.4–11.3%; specificity: no cross-reactivity was observed with bilirubin (30 mg/dL), triglycerides (19.7 mmol/L), hemoglobin (10 g/L), androstenedione (1000 μg/dL), cortisol (10,000 μg/dL), estradiol (5000 μg/dL), or testosterone (2000 μg/dL); reference ranges for women: 18–20 years: 51–321 μg/dL; 21–30 years: 18–391 μg/dL; 31–40 years: 23–266 μg/dL. Serum sex hormone-binding globulin (SHBG) concentrations were measured by chemiluminescent immunoassay (CLIA) using an original reagent kit on the Access 2 Immunoassay System (Beckman Coulter, Inc., Brea, CA, USA), in strict accordance with the manufacturer’s instructions. The method has an analytical sensitivity of 0.33 nmol/L and a reportable measuring range up to 180 nmol/L, as specified by the manufacturer. Intra-assay and inter-assay coefficients of variation (CVs) are generally <10%. Calibration was performed using manufacturer-provided calibrators traceable to internal standards.

The free androgen index (FAI) was calculated using the following formula: Testosterone (nmol/L) × 100/SHBG (nmol/L).

Serum leptin levels were quantified by a solid-phase human ELISA method using a commercial kit (DRG Instruments GmbH, Marburg, Germany) with the following characteristics: sensitivity: 0.2 ng/mL; intra-assay CV < 8.7%; inter-assay CV < 5.4%. Serum adiponectin concentrations were determined using a human ELISA kit (BioVendor—Laboratorní medicína a.s., Heidelberg, Germany) with the following features: sensitivity: 26 ng/mL; intra-assay CV < 5.9%; inter-assay CV < 7.0%. Visfatin concentrations were measured using an ELISA kit (catalog no. CSB-E08940h; Gentaur Molecular Products, Kampenhout, Belgium) with the following characteristics: sensitivity: 0.16 ng/mL; intra-assay CV 4.0–6.0%; inter-assay CV 8.0–12.0% and specificity: no detected cross-reactivity with similar proteins. Levels of serum resistin were measured by a competitive solid-phase human EIA kit (Phoenix Pharmaceuticals, Inc., Burlingame, CA, USA), characterized by sensitivity: 1.16 ng/mL; intra-assay CV < 14.0%; inter-assay CV < 5.0%. Serum TNF-α levels were determined using a solid-phase immunoassay (Human ELISA method DRG Instruments GmbH, Marburg, Germany), featuring the following characteristics: sensitivity: 3.0 pg/mL; intra-assay variation: CV%. Serum concentrations of IL-6 were measured using a solid-phase immunoassay Human ELISA method (DRG Instruments GmbH, Marburg, Germany), characterized by the following specifications: sensitivity: 2.0 pg/mL; intra-assay variation: CV% < 7.7.

For a more precise assessment of cardiovascular risk, the leptin-to-adiponectin ratio (L/A) and adiponectin-to-log 10 Resistin ratio (A/R) were calculated.

### 4.4. Statistical Analysis

All statistical analyses were conducted using SPSS software, version 21.0 (IBM Corp., Armonk, NY, USA) for Windows. The normality of data distribution was assessed using the Kolmogorov–Smirnov test. Resistin levels were log_10_-transformed prior to analysis to achieve a normal distribution. Data for normally distributed variables were presented as mean ± standard deviation (SD), whereas non-normally distributed variables were expressed as median and interquartile range. Between-group comparisons were performed using Student’s *t*-test or Mann–Whitney U test, as appropriate. Correlations were assessed using Pearson, Spearman, or Kendall’s tau coefficients, depending on the data type and distribution. Given the number of adipokines and metabolic variables tested, we applied [Bonferroni correction/FDR correction] for multiple comparisons. Both unadjusted and adjusted *p*-values are presented. For borderline results (0.05 < *p* < 0.10), findings are described as non-significant.

To examine the independent contributions of adiposity and insulin resistance measures to adipokine and metabolic outcomes, we applied multivariate general linear models (GLMs) adjusted for age, entering WHtR and HOMA-IR as continuous covariates. This approach allowed simultaneous testing of multiple dependent variables (adiponectin, leptin, resistin, visfatin, TNF-α, IL-6, atherogenic indices, and non-HDL-C), providing overall multivariate effects as well as univariate effects for individual outcomes. Stepwise linear regression models were employed to identify predictors of leptin levels and to examine the relationships between adipokines and insulin resistance. A two-tailed *p*-value < 0.05 was considered statistically significant. No imputation was performed for missing data.

## 5. Conclusions

This study reveals a distinct metabolic and inflammatory adipokine profile in women with PCOS, characterized by elevated levels of leptin, visfatin, and resistin, as well as reduced adiponectin. These alterations were associated with increased adiposity, impaired glucose metabolism, and adverse atherogenic indices, while lipid changes primarily affected triglycerides and AIP. Multivariate general linear model analyses further indicated that central adiposity, reflected by WHtR, rather than insulin resistance (HOMA-IR) per se, was the key independent determinant of adipokine alterations and atherogenic indices. Elevated TNF-α and the associations between resistin and IL-6 suggest intertwined adipokine–inflammatory signaling. Together, these findings underscore the potential of adipokines as integrated biomarkers of cardiometabolic risk in PCOS, highlighting that interventions targeting central fat distribution may improve metabolic, reproductive, and cardiovascular outcomes. Future longitudinal and mechanistic studies are warranted to validate these relationships and to explore phenotype-specific differences that may inform personalized management strategies.

## 6. Patents

The authors declare that no patents have resulted from this study.

## Figures and Tables

**Figure 1 ijms-26-09702-f001:**
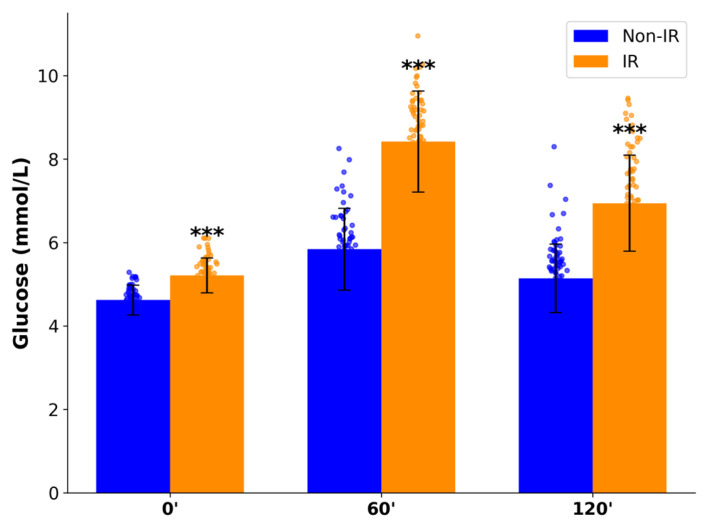
Glucose levels (mmol/L) at baseline and during OGTT in IR and non–IR women with PCOS. ***—*p* < 0.001.

**Figure 2 ijms-26-09702-f002:**
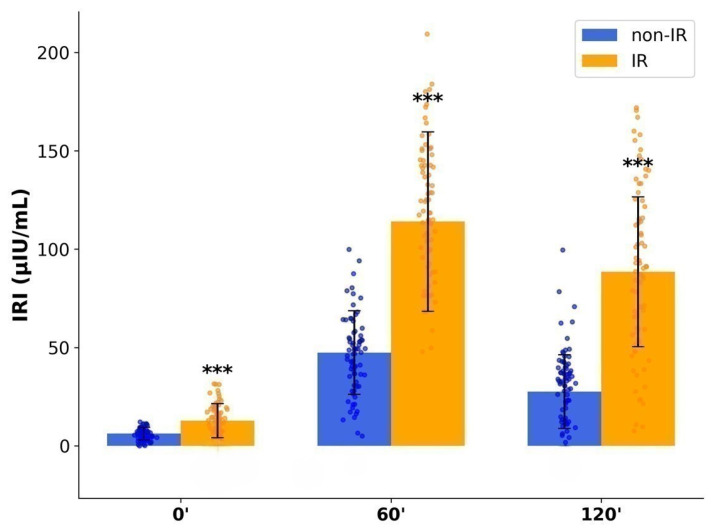
Insulin levels (μIU/mL) at baseline and during OGTT in IR and non–IR PCOS women. ***—*p* < 0.001.

**Figure 3 ijms-26-09702-f003:**
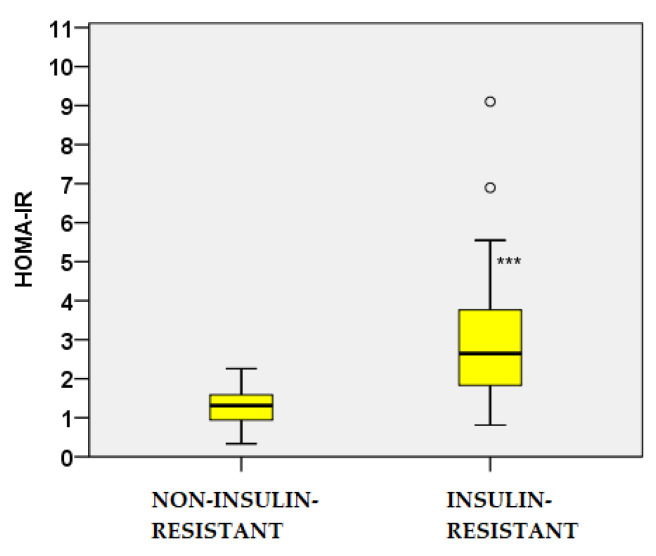
HOMA-IR values in IR and non–IR women with PCOS. ***—*p* < 0.001; circles represent outlier values outside the 1.5 interquartile range.

**Figure 4 ijms-26-09702-f004:**
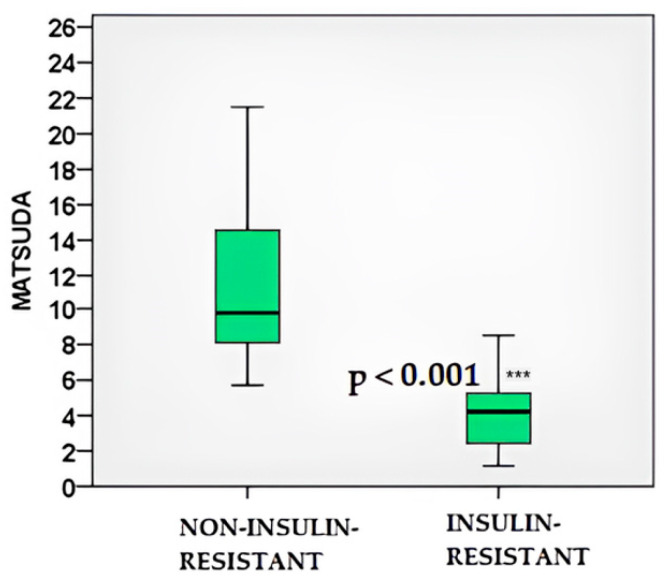
Matsuda index values in IR and non–IR women with PCOS. ***—*p* < 0.001.

**Figure 5 ijms-26-09702-f005:**
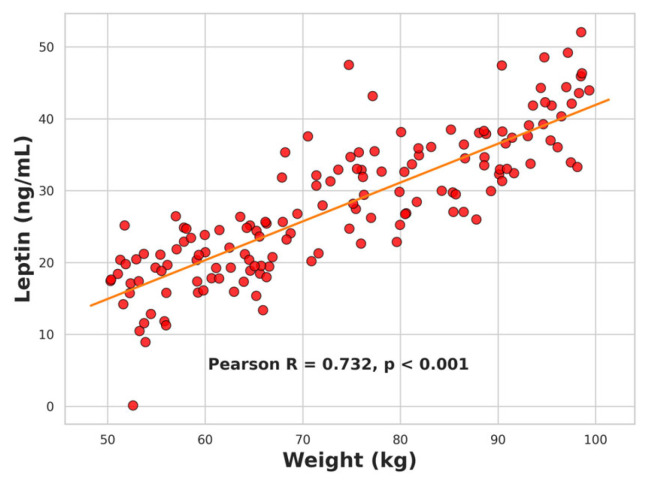
Scatter plot showing the positive correlation between leptin levels and body weight in the study population.

**Figure 6 ijms-26-09702-f006:**
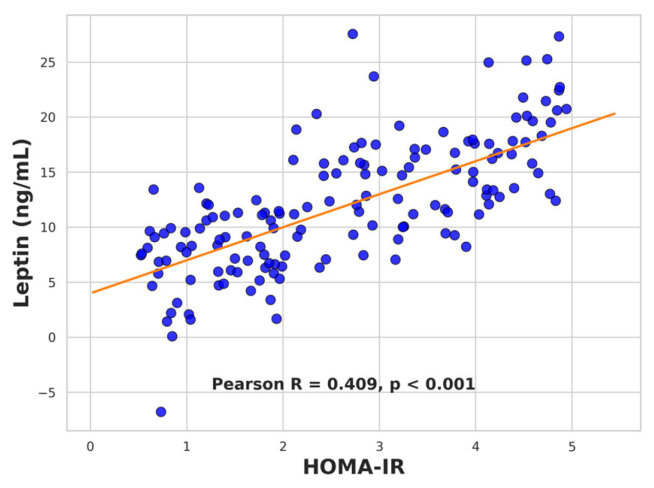
Scatter plot showing the positive correlation between leptin levels and HOMA-IR in the study population.

**Figure 7 ijms-26-09702-f007:**
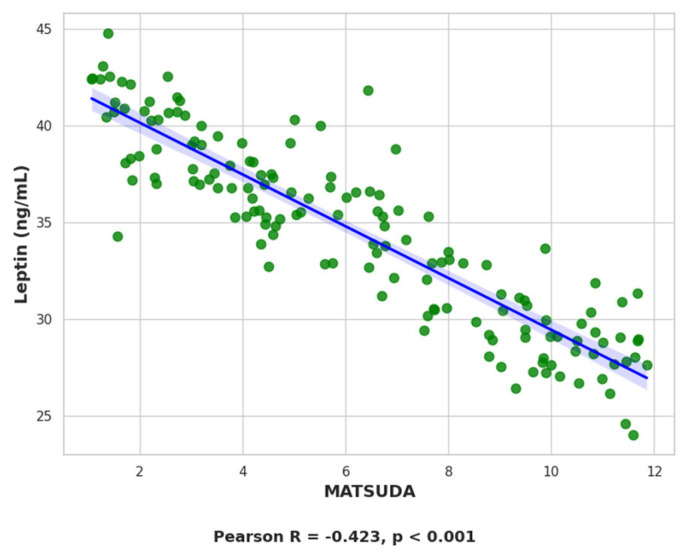
Scatter plot showing the negative correlation between leptin levels and Matsuda in the study population.

**Table 1 ijms-26-09702-t001:** Age and anthropometric parameters in the two studied groups of women with PCOS.

Parameters	Non-IR PCOS (*n* = 74)	IR PCOS (*n* = 76)
Age (years)	24.60 ± 4.53	24.03 ± 5.86 NS
Height (m)	1.67 ± 0.08	1.66 ± 0.05 NS
Weight (kg)	67.92 ± 15.94	78.08 ± 16.00 **
BMI (kg/m^2^)	24.31 ± 5.28	28.40 ± 5.56 **
Waist (cm)	79.16 ± 13.45	89.21 ± 16.17 **
WHtR	48.22 ± 8.30	53.72 ± 10.16 *

NS—not significant (*p* > 0.05); *—*p* < 0.05; **—*p* < 0.01.

**Table 2 ijms-26-09702-t002:** Adipokines in the two studied groups of PCOS women.

Adipokines	Non-IR PCOS (*n* = 74)	IR PCOS (*n* = 76)
Visfatin (ng/mL)	7.23 ± 3.76	14.05 ± 11.03 *
Leptin (ng/mL)	24.90 ± 18.37	39.56 ± 18.54 **
Adiponectin (mcg/mL)	14.70 ± 7.74	9.19 ± 4.53 **
Log10 Resistin (ng/mL)	0.67 ± 0.20	0.81 ± 0.23 *
IL-6 (pg/mL)	1.28 ± 1.07	1.29 ± 0.90 NS
TNF-α (pg/mL)	7.15 ± 5.63	9.80 ± 5.94 Δ *

Δ—Mann–Whitney U-test; NS—not significant, (*p* > 0.05); *—*p* < 0.05; **—*p* < 0.01.

**Table 3 ijms-26-09702-t003:** Lipid profile parameters and calculated atherogenic indices in the two study groups of women with PCOS.

Parameters	Non-IR PCOS (*n* = 74)	IR PCOS (*n* = 76)
TC (mmol/L)	4.48 ± 0.93	4.46 ± 0.86 NS
HDL-C (mmol/L)	1.37 ± 0.49	1.23 ± 0.28 NS
LDL-C (mmol/L)	2.78 ± 0.92	2.67 ± 0.84 NS
TG (mmol/L)	0.80 ± 0.30	1.22 ± 0.60 ***
Non-HDL-C	3.17 ± 0.94	3.18 ± 0.97 NS
AIP	−0.09 ± 0.14	0.03 ± 0.19 **
L/A	2.80 ± 2.15	5.28 ± 3.17 ***
A/R	21.78 ± 12.42	12.41 ± 7.48 ***

NS—not significant (*p* > 0.05); **—*p* < 0.01; ***—*p* < 0.001.

**Table 4 ijms-26-09702-t004:** Hormonal parameters and FAI in the two studied groups of women with PCOS.

Parameters	Non-IR PCOS (*n* = 74)	IR PCOS (*n* = 76)
Total testosterone (ng/mL)	0.66 ± 0.18	0.74 ± 0.19 *
SHBG (nmol/L)	42.26 ± 26.54	39.51 ± 23.28 NS
Androstenedione (ng/mL)	4.03 ± 1.98	3.28 ± 1.58 NS
DHEA-S (μg/dL)	282.51 ± 111.21	283.88 ± 101.21 NS
FAI	6.85 ± 4.87	11.25 ± 9.41 NS

NS—not significant (*p* > 0.05); *—*p* < 0.05.

**Table 5 ijms-26-09702-t005:** Multivariable linear model analysis of adipokines and inflammatory markers in women with PCOS.

Dependent Variable	Predictor	F	*p*-Value
Leptin	WHtR	6.98	0.013 *
	HOMA-IR	1.50	0.230 NS
Adiponectin	WHtR	2.57	0.118 NS
	HOMA-IR	0.46	0.503 NS
Log10 Resistin	WHtR	0.13	0.721 NS
	HOMA-IR	2.16	0.152 NS
Visfatin	WHtR	0.04	0.853 NS
	HOMA-IR	0.45	0.509 NS
TNF-α	WHtR	0.72	0.402 NS
	HOMA-IR	0.00	0.971 NS
IL-6	WHtR	0.16	0.689 NS
	HOMA-IR	0.66	0.421 NS

NS—not significant (*p* > 0.05); *—*p* < 0.05.

**Table 6 ijms-26-09702-t006:** Multivariable linear model analysis of atherogenic indices and non-HDL-C in women with PCOS.

Dependent Variable	Predictor	F	*p*-Value
**L/A**	WHtR	10.74	0.003 **
	HOMA-IR	0.35	0.560 NS
**A/R**	WHtR	1.44	0.239 NS
	HOMA-IR	1.35	0.254 NS
**Non-HDL-C**	WHtR	6.71	0.014 *
	HOMA-IR	0.91	0.347 NS
**AIP**	WHtR	10.97	0.002 **
	HOMA-IR	1.54	0.223 NS

NS—not significant (*p* > 0.05); *—*p* < 0.05; **—*p* < 0.01.

## Data Availability

The data that support the findings of this study are available from the corresponding author upon reasonable request. Due to ethical restrictions and participant confidentiality, the dataset is not publicly available.

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
