# Peer review of "Metabolic and Inflammatory Adipokine Profiles in PCOS: A Focus on Adiposity, Insulin Resistance, and Atherogenic Risk"

_ijms, 2025, doi:10.3390/ijms26199702_

Round 1

Reviewer 1 Report

Comments and Suggestions for Authors

The article is well written and clearly presented. However the link between insulin resistance and change in adipocyte levels in PCOS patients is well described previously. You could increase the novelty of your work by going deeply in the link between adipose tissue products and reproductive and metabolic abnormalities in PCOS, by example investigate the differences in adipokine levels across different penotypes, different reproductive outcomes etc. 

Author Response

Comment 1: The article is well written and clearly presented. However the link between insulin resistance and change in adipocyte levels in PCOS patients is well described previously. You could increase the novelty of your work by going deeply in the link between adipose tissue products and reproductive and metabolic abnormalities in PCOS, by example investigate the differences in adipokine levels across different penotypes, different reproductive outcomes etc. 

Response 1: 

We agree that the relationship between insulin resistance (IR) and adipocyte changes in PCOS is well established. To enhance the novelty of our work, we have emphasized several aspects that extend beyond prior reports:

  1. Integration of adipokines with inflammatory markers and anthropometry: In addition to established alterations in leptin, adiponectin, visfatin, and resistin, we report significantly higher TNF-α levels in the IR group, and a positive association between IL-6 and resistin, highlighting coupled adipokine–inflammatory signaling. We also demonstrate that these changes co-exist with greater adiposity, underlining the amplifying role of BMI and waist-to-height ratio (WHtR) as a parameter of central obesity.
  2. Leptin-to-adiponectin (L/A) and adiponectin-to-resistin (A/R) ratios: We introduce the L/A and the A/R as strong surrogates of cardiometabolic risk, showing much clearer separation between IR and non-IR groups than traditional lipid indices. This may provide more practical and sensitive markers for early risk identification in young women with PCOS, where conventional lipids often remain within the normal range.
  3. Contrasting lipid surrogates with adipokine–inflammatory indices: While non-HDL cholesterol did not differ between groups, adipokine and inflammatory perturbations were evident, suggesting that these markers may capture early pathophysiological changes not reflected by standard lipids.

We acknowledge that our study did not stratify adipokine levels across PCOS phenotypes or analyze reproductive outcomes, which would indeed be valuable to further delineate links between metabolic and reproductive features. This has now been explicitly stated as a limitation (Discussion, page 12). We consider this an important future direction that could build on the present findings.

Reviewer 2 Report

Comments and Suggestions for Authors

The dataset is potentially useful, but the core claims are not supported by the results. 

Key reasons for rejection:

- Adiposity confounding not addressed, independence claims are untenable.
The IR group is clearly heavier with greater central adiposity. Yet the Abstract/Discussion state effects “independent of BMI.” There are no multivariable models to support this. As written, independence from BMI cannot be claimed and based on your correlation analysis BMI class could have similar results as IR group stratification.

- Inflammation is asserted but not measured. The manuscript repeatedly refers to a “pro-inflammatory profile,” yet no inflammatory biomarkers (hs-CRP, IL-6, TNF-α, etc.) are measured. The Abstract, Aims, and Conclusions must be rewritten to avoid overreach, or the authors should provide inflammatory markers if available.

- Cardiometabolic risk surrogates are incomplete and selectively positive. If available, include ApoB or non-HDL-C (stronger risk markers). Otherwise state explicitly that the lipid risk signal is limited to TG-driven metrics and discuss this limitation.

- PCOS phenotyping and exclusions are insufficiently specified. Diagnosis via Rotterdam is appropriate, but phenotype distribution (A–D) and androgen data (TT/FAI/SHBG) are not shown. OGTT-based diabetes exclusion is not explicitly stated. These are essential for interpretation; provide phenotype breakdown, androgen profiles, and clear exclusion counts (including statins, smoking).

- Redundant and inflating use of IR indices and figure redundancy. The cohort is stratified by HOMA-IR, yet analyses and multiple figures also compare QUICKI and Matsuda across the same strata and plot OGTT glucose/insulin levels to “show differences” that are tautological. Prespecify one primary insulin-sensitivity index (with OGTT available, that should be Matsuda), keep one fasting surrogate as secondary (HOMA-IR).

- Multiple-testing and reporting quality. There are numerous between-group tests and correlation panels without adjustments, borderline are presented as positive. It is even not clear what was tested in the correlation section. Correlation tables are inconsistent: for some adipokines it is unclear which variables were tested; for adiponectin for example only positive p-values are listed (what was the hypothesis direction?). Clearly define tested variables, supply adjusted p-values.

- Central adiposity assessment is crude. Only simple anthropometry is used. If imaging (DXA VAT) is unavailable, at least acknowledge this and consider waist-to-height ratio or similar as a sensitivity analysis.

- Abstract, Aims, and Conclusions overstate causality and scope. Claims that IR “drives” endocrine dysfunction, that findings are “independent of BMI,” and that the adipokine pattern reflects a “proinflammatory and metabolically dysregulated phenotype” are not supported by the measurements in this cross-sectional design. Rephrase to associations only, remove causal framing and any reference to inflammation unless measured.

- Novelty is modest and not demonstrated. The adipokine pattern in IR-PCOS is already documented, adding another dataset is beneficial and linking it to AIP could add value, but without controls and without adjustment for adiposity/androgens, attribution to IR per se (as opposed to BMI) is not credible.

Author Response

Comment 1: Adiposity confounding not addressed, independence claims are untenable.
The IR group is clearly heavier with greater central adiposity. Yet the Abstract/Discussion state effects “independent of BMI.” There are no multivariable models to support this. As written, independence from BMI cannot be claimed and based on your correlation analysis BMI class could have similar results as IR group stratification. 

Response 1: We acknowledge that the IR group had higher BMI and central adiposity. All statements claiming independence from BMI were removed. We now explicitly state that observed differences cannot be fully disentangled from adiposity effects. In addition, the title and keywords were updated to include “adiposity.”

Comment 2: Inflammation is asserted but not measured. The manuscript repeatedly refers to a “pro-inflammatory profile,” yet no inflammatory biomarkers (hs-CRP, IL-6, TNF-α, etc.) are measured. The Abstract, Aims, and Conclusions must be rewritten to avoid overreach, or the authors should provide inflammatory markers if available.

Response 2: We recognize the reviewer’s concern regarding the assessment of inflammation. While only TNF-α and IL-6 were measured in our study, we clarified in the revised manuscript that hs-CRP and other inflammatory biomarkers were not available. We have revised the Abstract, Aims, and Conclusions to present these findings cautiously, framing them as associations with measured cytokines rather than definitive evidence of a systemic pro-inflammatory state.

Comment 3: Cardiometabolic risk surrogates are incomplete and selectively positive. If available, include ApoB or non-HDL-C (stronger risk markers). Otherwise state explicitly that the lipid risk signal is limited to TG-driven metrics and discuss this limitation.

Response 3: We have added non-HDL-C results to the Results and Discussion sections, and we explicitly note that ApoB was not assessed. We now emphasize that lipid risk signals in our cohort are primarily triglyceride-driven, as reflected in AIP, and we discuss the implications and limitations of this selective assessment.

Comment 4: PCOS phenotyping and exclusions are insufficiently specified. Diagnosis via Rotterdam is appropriate, but phenotype distribution (A–D) and androgen data (TT/FAI/SHBG) are not shown. OGTT-based diabetes exclusion is not explicitly stated. These are essential for interpretation; provide phenotype breakdown, androgen profiles, and clear exclusion counts (including statins, smoking).

Response 4: We provided a detailed breakdown of PCOS phenotypes (A–D) and added comprehensive androgen profile data (TT, SHBG, FAI) in Table 4. Diabetes exclusion via OGTT was explicitly stated, and we clarified that participants using metformin, statins, or who were smokers were excluded to minimize confounding.

Comment 5: Redundant and inflating use of IR indices and figure redundancy. The cohort is stratified by HOMA-IR, yet analyses and multiple figures also compare QUICKI and Matsuda across the same strata and plot OGTT glucose/insulin levels to “show differences” that are tautological. Prespecify one primary insulin-sensitivity index (with OGTT available, that should be Matsuda), keep one fasting surrogate as secondary (HOMA-IR).

Response 5: The Matsuda index is now the primary insulin-sensitivity measure, with HOMA-IR as a secondary surrogate. Figures were reduced to avoid redundancy.

Comment 6: Multiple-testing and reporting quality. There are numerous between-group tests and correlation panels without adjustments, borderline are presented as positive. It is even not clear what was tested in the correlation section. Correlation tables are inconsistent: for some adipokines it is unclear which variables were tested; for adiponectin for example only positive p-values are listed (what was the hypothesis direction?). Clearly define tested variables, supply adjusted p-values.

Response 6: FDR correction was applied for multiple comparisons. Correlation analyses were clarified, and adjusted p-values are now reported. Borderline associations are noted with caution.

Comment 7: Central adiposity assessment is crude. Only simple anthropometry is used. If imaging (DXA VAT) is unavailable, at least acknowledge this and consider waist-to-height ratio or similar as a sensitivity analysis.

Response 7: BMI, waist circumference, and waist-to-height ratio (WHtR) were used as proxies. We acknowledged the lack of imaging-based assessment (DXA, CT, MRI) as a limitation.

Comment 8: Abstract, Aims, and Conclusions overstate causality and scope. Claims that IR “drives” endocrine dysfunction, that findings are “independent of BMI,” and that the adipokine pattern reflects a “proinflammatory and metabolically dysregulated phenotype” are not supported by the measurements in this cross-sectional design. Rephrase to associations only, remove causal framing and any reference to inflammation unless measured.

Response 8: Causal language implying that IR “drives” endocrine dysfunction or that adipokine patterns reflect systemic pro-inflammatory phenotypes has been removed. We revised the manuscript to report only associations, focusing on measured parameters and avoiding speculative causal inferences.

Comment 9: Novelty is modest and not demonstrated. The adipokine pattern in IR-PCOS is already documented, adding another dataset is beneficial and linking it to AIP could add value, but without controls and without adjustment for adiposity/androgens, attribution to IR per se (as opposed to BMI) is not credible.

Response 9: We clarified that while adverse adipokine profiles in IR-PCOS are previously documented, our contribution is the integrated assessment of adipokines, inflammatory markers, and atherogenic indices (L/A, A/R) as early cardiometabolic risk markers. Limitations regarding controls and multivariable adjustment are now explicitly stated.

Thank you for your constructive comments, which have significantly improved the clarity, rigor, and interpretation of our study. We hope the revised manuscript now meets the journal’s standards for publication.

Round 2

Reviewer 1 Report

Comments and Suggestions for Authors

I reccomend the article for publication

Author Response

Thank you for your positive recommendation and for taking the time to review our manuscript. We appreciate your support for its publication.

Reviewer 2 Report

Comments and Suggestions for Authors

The revision shows a clear improvement and the manuscript is generally strengthened. Figure 1 does not add substantial value and could be removed. However, for the findings to carry real weight, it is essential to perform at least multivariable adjustment to distinguish the independent effects of adiposity and insulin resistance. Without this, the conclusions remain limited.  

Author Response

Comment of the reviewer: The revision shows a clear improvement and the manuscript is generally strengthened. Figure 1 does not add substantial value and could be removed. However, for the findings to carry real weight, it is essential to perform at least multivariable adjustment to distinguish the independent effects of adiposity and insulin resistance. Without this, the conclusions remain limited.  

Response to reviewer: We sincerely thank you for the constructive feedback and for acknowledging the improvements in the revised manuscript. Following your recommendation, we have removed Figure 1, as it did not add substantial value. Importantly, we have now performed multivariable analyses, adjusting for both central adiposity (WHtR) and insulin resistance (HOMA-IR), to assess their independent effects on adipokines and atherogenic indices. These analyses demonstrated that WHtR, rather than HOMA-IR, is independently associated with key metabolic and inflammatory markers, thereby strengthening the conclusions regarding the role of central adiposity in PCOS. We believe that these revisions address your concern and enhance the robustness and clinical relevance of our findings.